# A Review of the Hydraulic Performance of Permeable Reactive Barriers Based on Granular Zero Valent Iron

Stefania Bilardi *, Paolo Salvatore Calabrò and Nicola Moraci

Department DICEAM, Via Graziella, Mediterranea University of Reggio Calabria, Loc. Feo di Vito, 89122 Reggio Calabria, Italy
* Correspondence: stefania.bilardi@unirc.it

**Abstract:** Permeable reactive barriers (PRBs) based on the use of zero valent iron (ZVI) represent an efficient technology for the remediation of contaminated groundwater, but the literature evidences "failures", often linked to the difficulty of fully understanding the long-term performance of ZVI-based PRBs in terms of their hydraulic behavior. The aim of this paper is to provide an overview of the long-term hydraulic behavior of PRBs composed of ZVI mixed with other reactive or inert materials. The literature on the hydraulic performance of ZVI-based PRBs in full-scale applications, on long-term laboratory testing and on related mathematical modeling was thoroughly analyzed. The outcomes of this review include an in-depth analysis of factors influencing the long-term behavior of ZVI-based PRBs (i.e., reactive medium, contamination and the geotechnical, geochemical and hydrogeological characteristics of the aquifer) and a critical revision of the laboratory procedures aimed at investigating their hydraulic performance. The analysis clearly shows that admixing ZVI with nonexpansive granular materials is the most suitable choice for obtaining a long-term hydraulically efficient PRB. Finally, the paper summarizes a procedure for the correct hydraulic design of ZVI-based PRBs and outlines that research should aim at developing numerical models able to couple PRBs' hydraulic and reactive behaviors.

**Keywords:** clogging; column tests; granular mixtures; heavy metals; hydraulic conductivity; iron corrosion; mathematical models; PRB design

## 1. Introduction

Research in the field of in situ groundwater remediation has recently focused on innovative technologies that are highly efficient and environmentally sustainable, particularly those that are capable of making contaminated groundwater resources usable again. These technologies include permeable reactive barriers (PRBs), which are composed of granular reactive materials. When contaminated groundwater flows through a PRB due to the effect of the natural gradient, contaminants that are present in the groundwater are degraded and/or immobilized because of the processes that are activated by the PRB filling material [1,2]. The choice of reactive granular filling represents a key point in the design of PRBs and requires the consideration of both the removal efficiency and hydraulic conductivity in the long term. These aspects have generally been neglected in the scientific literature.

Some examples of granular reactive media that have been used in PRBs are limestone, zero-valent iron (ZVI), zeolite, compost, hydroxyapatite, granular activated carbon (GAC), sodium dithionite, blast furnace slag or mixtures of materials, such as ZVI and peat or ZVI and iron shavings [1,3,4]. These reactive materials induce pollutant immobilization/degradation based on several mechanisms, such as precipitation, coprecipitation, adsorption or redox reactions [2,5].

The use of other low-cost reactive media that are derived from industrial byproducts or waste materials can further minimize the impacts of PRBs on the ecosystem, not only thanks to groundwater reclamation but also reductions in the use of virgin resources.

Some examples of these materials include crushed cocoa shells [6], compost [7], recycled concrete [8], cement kiln dust [9], coal fly ash [10], lapillus [11] and sand coated with humic acids extracted from sewage sludge [12]. The use of these materials is subject to accurate characterization in order to avoid the release of contaminants.

The most commonly used reactive material in chemical–physical barriers is ZVI since it is able to treat water that has been contaminated with a large variety of pollutants, including chlorinated organic solvents, heavy metals, radionuclides or mixed contamination (for example, heavy metals and chlorinated solvents) [1,13,14]. ZVI is able to activate several removal mechanisms, such as coprecipitation (i.e., dissolved species are mechanically entrapped in the matrices of oxyhydroxides during their precipitation), contaminant reduction mediated by iron corrosion products (e.g., $Fe^{II}$ and $H/H_2$) and adsorption on iron oxides [5,15].

According to the most important reviews on ZVI-based PRBs, the critical issues that have emerged over the last twenty years include the following:

- the life span of ZVI [2,4,16–22];
- the necessity for and relevance of an in-depth knowledge of hydrogeology and the nature of contaminated plumes [13,16,17];
- the necessity for detailed and comprehensive field monitoring to evaluate the effectiveness of the PRBs [2,23];
- the necessity for a better understanding of the removal mechanisms activated by ZVI-based aqueous systems [22,24–27].

These issues have arisen from the awareness that although PRBs generally perform well after over a decade of operation [28,29], their long-term performance is still not well understood, especially in terms of hydraulic conductivity, (this statement has remained unchanged from 2007 until now [23]). For this reason, this review aimed to investigate the hydraulic behavior of PRBs composed of ZVI in light of the most recent findings. The first part of this paper provides an introduction to the main requirements for PRBs and the possible mechanisms of interaction between contaminated groundwater and ZVI. Then, we present our examination of cases in which reductions in the hydraulic conductivity of PRBs influenced their correct operation by analyzing their performance in (i) full-scale applications, (ii) laboratory tests and (iii) mathematical models. Furthermore, we also discuss our analysis of the strategies that have been used to improve the hydraulic behavior of ZVI systems. This study allowed us to develop useful suggestions for the correct design of hydraulically efficient ZVI-based PRBs and ideas for research into new granular mixtures that could be capable of optimizing ZVI use.

## 2. Main Requirements for a PRB

### 2.1. PRB Configuration

The placement of a reactive material in the subsoil downstream of a contaminant source defines the precise configuration of a PRB (Figure 1). The most simple configuration, which is suitable for almost uniform aquifers, is "horizontal", which is obtained by arranging the reactive medium in a perpendicular direction to the flow of groundwater [30]. In the funnel and gate configuration, elements with lower permeability "funnel" the contaminated flow directly toward the reactive medium "gate", whereas in the caisson configuration, the reactive material is placed in a caisson and the contaminated plume flows vertically upward through the caisson [31,32]. Compared to the horizontal configuration, the funnels that are placed upstream of the reactive and permeable gates cause flows to mix, resulting in lower variability in influent and effluent concentrations and more efficient use of the reactive medium [33]. The caisson configuration is the most expensive type of PRB but is appropriate when multiple zones comprising several reactive media are needed or when the periodic rejuvenation or replacement of the reactive medium is required [33].

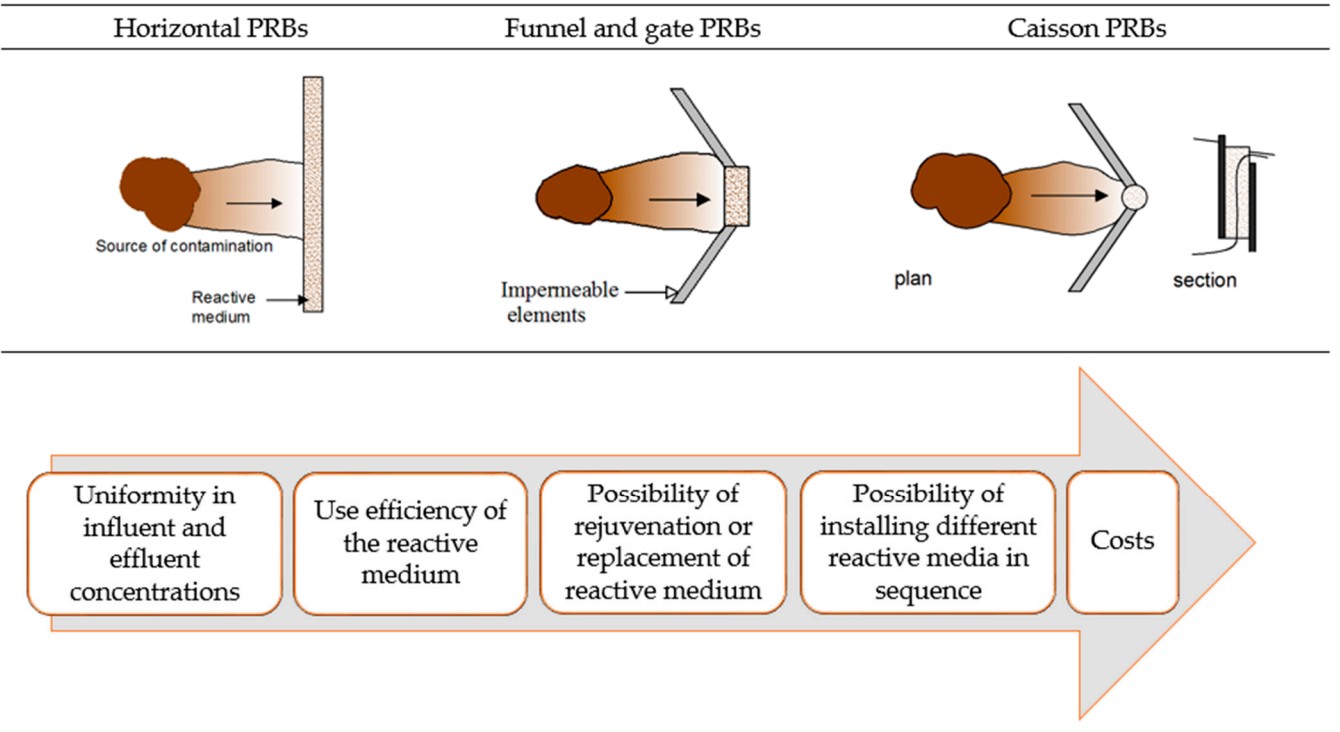

**Figure 1.** PRB configurations and their main characteristics.

### 2.2. Sustainability

In addition to the most appropriate configuration, the choice of reactive medium ensures the correct operation of PRBs. The first studies on PRBs recommended some simple rules for selecting the reactive material. In addition to efficiently removing contaminants, the material should also be compatible with the surrounding environment, i.e., the material must not cause adverse chemical reactions with groundwater or represent a possible source of secondary contamination. These requirements essentially refer to the concept of sustainability, which has taken on a wider meaning over the years. In particular, the concept of sustainability can refer to the type of reactive material (for example, the use of waste materials or byproducts) or the overall operation of the remediation technology (i.e., sustainable remediation technology).

In the first case, the use of waste materials gives new life to materials, as long as their use does not damage the environment or remediation operators [34–36]. In this case, as with for all materials used in PRBs, their long-term efficiency must be carefully studied [37,38]. A reactive medium that is exhausted quickly or that loses its permeability may not be sustainable, especially if it has to be replaced several times, as its operation and subsequent disposal in a landfill can release significant negative emissions into the environment (e.g., greenhouse gases).

With reference to the operation of a remediation technology as a whole, sustainable remediation takes into account (i) the environmental, economic and social impacts of a technology or project and (ii) the need to involve stakeholders to guide decisions in a shared way. The precise definition of the sustainability of a remediation process involves the use of qualitative, semiquantitative and quantitative indicators [39–41]. A triple bottom line life cycle sustainability assessment allows for the quantification of the key indicators of the environmental, economic and social impacts across the project's life cycle using appropriate metrics. Some examples of quantifiable environmental impact indicators that are relevant for the sustainability assessment of remedial alternatives are energy and fossil fuel consumption, waste generation and the depletion of natural resources. With reference to the economic impacts, the indicators include the direct costs (materials,

labor, equipment, transport, etc.) and indirect costs (social costs of $CO_2$, $CH_4$ and $N_2O$, etc.). Finally, considering the social impacts, some examples of indicators include worker disturbance, community safety, lifestyle and economic development [42].

From an initial examination of the possible impacts of PRBs in the three aforementioned areas, it can be stated that this technology is characterized by a low environmental impact because it does not require energy consumption for its operation, it does not produce visible impacts on the ground (as the intervention is underground), it only requires the removal of a limited amount of subsoil and it allows for the use of water resources. Additionally, since the sites can still be used during remediation and the technology does not incur high construction costs, PRBs generate low social and economic impacts compared to other possible technologies (Figure 2).

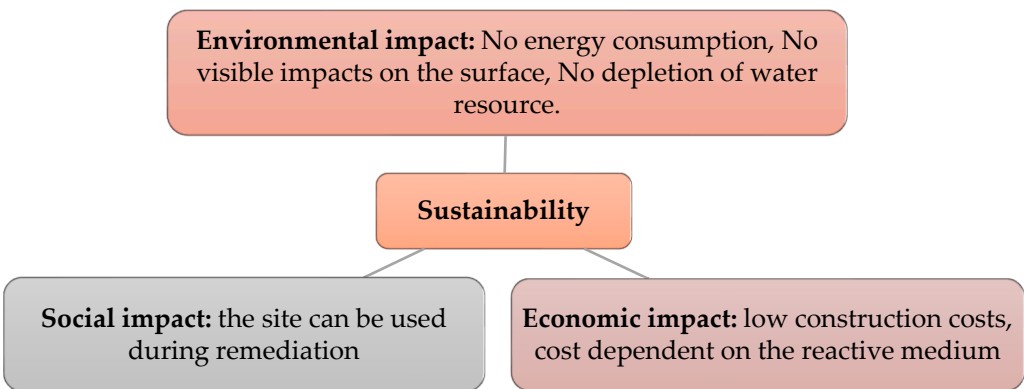

**Figure 2.** The environmental, economic and social impacts of PRBs.

In addition to the concept of sustainable remediation, the design of remediation processes should also incorporate the concept of resilience due to the recent impacts of climate change; however, this concept is new within this field and has not yet been widely explored by the scientific community.

Resilience is a measure of the ability of a system to absorb and adapt to the impacts of sudden changes in environmental conditions. With reference to the concept of resilience, remediation projects should consider the possible impacts of climate change and their ability to adapt to them. As with most remediation technologies, PRBs also require several years of operation to achieve remediation goals. For this reason, it may be important to take resilience to climate change into account during PRB design (e.g., fluctuations in the water tables of aquifers) in order to avoid unwanted environmental impacts [42].

Again, a precise knowledge of the long-term hydraulic performance of the barrier is of critical importance, especially when evaluating the requirement for fill material replacement.

### 2.3. Reactive Medium Selection

The selection of the most suitable reactive medium for PRB construction depends on two fundamental properties: reactivity toward contaminants and grain size distribution (GSD). It is preferable for the material to remain reactive for the entire period necessary for the decontamination of the contaminated plume in order to avoid the need for replacement. Additionally, the particle dimensions are fundamental to guaranteeing not only the reactivity of the material but also its long-term hydraulic efficiency. Usually, smaller particles have larger specific surface areas but present lower values of hydraulic conductivity.

Granular ZVI is available with different GSDs and each GSD has its own reactive and hydraulic behavior (according to the installation procedures). For PRBs constructed using excavation-based methods (in which the iron is placed directly into an excavation), the grain size range that is typically used is 2.0–0.25 mm [13]. Although this aspect has often been neglected in the literature, it is of fundamental importance to define the actual particle

size of the material investigated during laboratory experimentations or used in situ and not just the particle size range, as has been the case in most of the literature [38].

The reactive media that are installed in aquifers must fulfill a filter function with respect to the surrounding base soil. In fact, the filters must satisfy the following three main design criteria: internal stability, retention and permeability [43,44]. Internal stability is the ability of a filter to prevent the loss of its own small particles due to disturbing forces, such as seepage.

Regarding the retention and permeability criteria, under the dragging force of the groundwater flow, the filter (i.e., the material constituting the PRB) must be able to retain loose soil particles (retention criterion) to avoid particle clogging and allow seepage, thereby avoiding the development of high internal pore pressure (permeability criterion) [45].

The mandatory and desirable properties of reactive materials used in PRBs are summarized in Table 1. In particular, for the reasons explained above, it is mandatory to choose a reactive material that is internally stable (as described in detail in the fifth paragraph) and has a grain size distribution that is compatible with that of the aquifer. The chosen reactive material must be reactive toward contaminants and must not generate adverse chemical reactions. Other characteristics that are not strictly necessary but are desirable include the use of cheap materials or byproducts; for example, the production of granular ZVI is cheap and ZVI occurs as a byproduct of pig iron production [46].

**Table 1.** The required properties of reactive materials in PRBs.

| | |
|---|---|
| Grain Size Distribution (GSD) | Internally stable material (M) |
| | Compliance with filter design criteria (M) |
| | Readily available (D) |
| | By products (D) |
| | Low or moderate costs (D) |
| Chemical Composition | Reactivity toward contaminants (M) |
| | No generation of adverse chemical reactions (M) |
| | Long-term reactivity (D) |
| | Renewable after exhaustion (D) |
| Coefficient of Permeability | Permeability over time (M) |

Note: M, mandatory; D, desirable.

The selection of reactive media that are potentially suitable for the removal of one or more contaminants at certain concentrations is usually carried out via cheap and expeditious batch tests. These tests consist of putting each potential reactive medium in contact with a contaminated solution in tubes (one for each sampling time) at a given solid/liquid ratio. The tubes are usually mixed (e.g., using a rotary shaker) and withdrawn for analysis at preset time intervals. The reactive medium with the highest potential reduces the concentration of the pollutant faster and more effectively than the others.

Since the differences between experimental and field conditions are extensive, equilibrium isotherms help to calculate the adsorption capacity of materials, depending on the specific surface, grain size distribution (GSD), solution type, contaminant concentration and adsorbent dose or solid/liquid ratio [38]. However, batch experiments are not adequate for studying the real operating conditions of reactive media (i.e., the exhaustion of intrinsic reactivity, the possible loss of permeability and the hydraulic conditions); therefore, it is necessary to carry out more reliable tests (as described in Section 4.1) that can simulate the behavior of real PRBs in more detail [13].

## 3. Long-Term Hydraulic Behavior of PRBs

In order to achieve their remediation goals, PRBs must fulfill two functions: (i) the interception of contaminated groundwater plumes and (ii) reductions in contaminant concentrations to below regulatory limits or remediation goals [13,47].

To accomplish the first function, the hydraulic conductivity of a PRB must be greater than that of the aquifer by at least one order of magnitude [45]. To achieve the second function, the necessary residence time (i.e., the contact time between the groundwater and the reactive medium) must be guaranteed so that the desired chemical, physical, biological or mixed removal mechanisms can take place, according to the relevant kinetic conditions [13].

As already mentioned, two possible operational limits should be taken into account in PRB designs. The first is the reduction in the reactivity of the filling material, which does not allow for the achievement of remediation goals downstream of the barrier. The second is the reduction in hydraulic conductivity, which obstructs the aquifer flow through the barrier, causing the possible circumvention of the contaminated plume or significant increases in internal pore pressure [22,48,49].

In order to study these operational limits, it is necessary to understand the variables that regulate the transportation of pollutants through PRBs and the possible causes of variations in transportation properties. These variables are related to the reactive medium (type, grain size, density and activable removal mechanisms), contamination (type of contaminant(s) and initial concentration), the geotechnical properties of the aquifer (grain size and permeability), the geochemical characteristics of the aquifer (pH, dissolved oxygen and the presence of chemical species, such as Ca, Mg, Na, $SO_4^{2-}$, etc.) and the hydrogeological characteristics of the aquifer (hydraulic gradient and hydraulic conductivity).

Laboratory studies proposing new reactive materials (e.g., compost, peat, sawdust, ground rubber, leaf litter, limestone, zeolites, bone char, apatite (clinoptilolite), bauxite, activated alumina, wheat straw, softwood and sand and maize cobs [4]) have often neglected the hydraulic/geotechnical aspects, such as the grain size distribution curve of the reactive materials, and/or the hydraulic aspects, such as long-term hydraulic behavior [38,45,49,50]. This clearly demonstrates the importance of this issue for the potential real-world applicability of any proposed material.

### 3.1. Mechanisms of Interaction between Contaminated Groundwater and ZVI

When ZVI interacts with groundwater flow, a series of physical and chemical processes occur, some of which allow contaminant removal while others are side processes that are sometimes undesired. The possible mechanisms of the contaminant removal process that is activated by ZVI in water are chemisorption, electrostatic physisorption, coprecipitation and size exclusion [51,52]. The extent to which these processes play roles in groundwater remediation depends on the interactions between the contaminant and the iron species, which are highly pH-dependent [52]. Spontaneous electrochemical processes involve chemical species whose oxidation reduction potential is slightly higher than that of iron ($E^0_{Fe2+/Fe0}$ = −0.44 V). The oxidation of ZVI to ferrous iron provides the driving force that reduces many redox-sensitive contaminants [52]. $H_2O$ or $H^+$ are also oxidizing agents for ZVI under mainly anaerobic natural conditions in groundwater (the electrode potential for the redox couple $H^+/H_2$ is 0.00 V). Therefore, when ZVI is immersed in contaminated or uncontaminated water, it corrodes to form $H/H_2$ and $Fe^{II}$ (and mixed $Fe^{II}/Fe^{III}$) species, which are stand-alone reducing agents and could contribute to contaminant removal. The oxidation of iron in water creates iron corrosion products (e.g., $Fe_2O_3$, FeOOH and green rust), which play significant roles in terms of remediation by means of chemisorption and coprecipitation processes [22,52,53].

The production of the iron corrosion products has the most important impact on barrier porosity since they occupy a volume that is 2 (for $Fe_3O_4$) to 6.4 (for $Fe(OH)_3 \cdot 3H_2O$) times greater than the volume of corroded iron [54]. This issue has also been neglected for almost the first two decades of research into ZVI-based PRBs.

Even the gas produced (i.e., $H_2$ under anaerobic conditions) and blocked in the pores of the filter due to the lack of proper ventilation can contribute to reductions in hydraulic conductivity in the long term [49,55].

The pH increase observed during iron corrosion can promote precipitation. This phenomenon occurs when a dissolved contaminant precipitates into a solid form and is immobilized inside the barrier. This phenomenon also involves mineral species that are present in the aquifer (for example, the formation of calcium or iron carbonates), which could cause a reduction in the hydraulic conductivity of the barrier.

Degradation is the process of the chemical or biological decomposition of a pollutant, which involves its transformation into a less toxic or harmless form; for example, organic materials/substrates can enhance the growth and activity of autochthonous or inoculated micro-organisms and facilitate the biodegradation of contaminants or the chemical reduction of chlorinated compounds by ZVI. In these compounds, chlorine atoms are replaced by hydrogen atoms as the molecule is reduced to ethylene, which can ultimately be metabolized into carbon dioxide and water by aerobic micro-organisms [56]. In the case of biological activity, biofilm growth or biocorrosion [57] are possible causes of reductions in porosity. Finally, if a barrier is not designed according to filter criteria [45], the retention of fine particles from upstream soil in the PRB pores can cause particle clogging.

As hydraulic conductivity increases, the reactivity of the filling material declines over time due to different causes. One cause is the progressive reduction in ZVI mass and its reactive surface due to the macroscopic dissolution of the metal. Another cause is the reduction in the iron's ability to generate new corrosion products, which are potential sites for contaminant adsorption or could help with coprecipitation [43,51,58]. Moreover, if the ZVI grain is passivated (i.e., covered with nonconductive corrosion products), its ability to participate in contaminant removal is impaired [59].

Thus, the reactive and hydraulic behavior of ZVI-based PRBs depends mainly on the iron corrosion rate, since its corrosion products are involved in both the removal of pollutants and the reduction in ZVI hydraulic conductivity. The iron corrosion rate, which depends on the intrinsic reactivity of the ZVI and the in situ conditions [60–64], allows for the activation of different chemical reactions that can affect the hydraulic behavior of ZVI-based PRBs (Figure 3). Due to the large number of variables, the study of the hydraulic conductivity behavior of ZVI-based PRBs is a complex phenomenon.

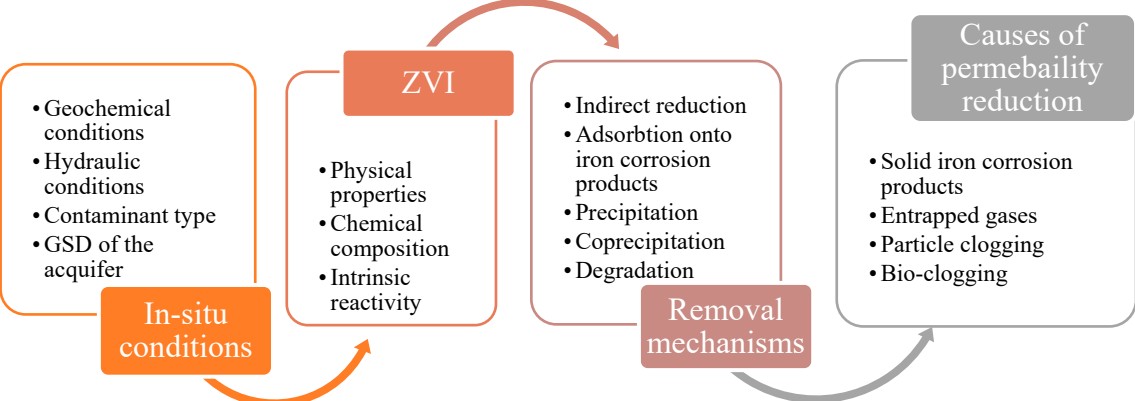

**Figure 3.** The factors that influence the long-term behavior of ZVI-based PRBs.

According to Hu et al. (2020) [22], the design of efficient and sustainable systems requires answering the question of how long iron corrosion products can be generated to satisfactorily treat contaminated groundwater while maintaining reasonable hydraulic conductivity. This study topic has emerged from the need to assess long-term iron corrosion rates as precisely as possible. For this purpose, over the last few years, different methods have been proposed to delineate the intrinsic reactivity of ZVI. These methods include the methylene blue (MB) method [65], $H_2$ evolution [66], iron dissolution in EDTA [67] and iron dissolution in 1,10-phenanthroline (i.e., the Phen test) [68]. Among these methods, the MB method can be reliably used as a reactive tracer for the semiquantitative characterization of the extent of iron corrosion. This method is based on the ability of sand to adsorb MB,

which is reduced when iron oxides cover its surface. Intrinsic reactivity can be indirectly assessed from the extent to which the iron corrosion products limit the adsorption of MB to the sand [65].

Another set of possible useful tools for characterizing the other physical characteristics of ZVI, such as morphology and specific surface area, were summarized in Li et al. (2019) [69].

### 3.2. Hydraulic Behavior of ZVI-Based PRBs Derived from Real-World Applications

Monitoring data on the hydraulic behavior of full-scale PRBs has outlined the presence of differing performances (Table 2). The first column of Table 2 shows the barrier filling reactive medium, the configuration and the treated contaminants. Other information contained in Table 2 includes the year of installation and the monitoring years, the site where the PRB was installed and the hydraulic behavior observed during PRB operations.

Good PRB performance was observed after 5 years of operation by O'Hannesin and Gillham [28] and after 15 years by Wilkin et al. [29], which demonstrated that the presence of mineral precipitates, such as calcium carbonate or iron oxides and sulfides, observed near the upstream PRB-aquifer interface did not significantly influence the hydraulic behavior of the PRB. However, in other cases, reductions in the hydraulic conductivity of the PRB [13,70,71] or the inefficient capture of contaminated groundwater [72] influenced PRB efficiency. The presence of mineral precipitates, such as calcium and iron carbonates, was mostly observed at the barrier inlet and, in some cases [73,74], cemented areas. Finally, the hydraulic performance of a PRB could also depend on the construction method, as in the case of the PRB installed in Nebraska [13,75]. According to that study, a possible reason for the early loss of PRB hydraulic conductivity was the uneven degradation of guar gum slurry, which could have penetrated the upgradient aquifer during construction, thereby promoting excessive microbial activity and sulfide precipitation. Guar gum is a biopolymer slurry that allows trenches to remain open during filling with reactive media.

**Table 2.** In situ observations of PRB hydraulic behavior.

| Description | Year Installed (Monitoring Years) | Site | Observations | References |
|---|---|---|---|---|
| ZVI/sand (22:78 w.r.) FG CS | 1991 (5 years) | Borden, Ontario | The presence of calcium carbonate near the upstream barrier-aquifer interface and the maintenance of the effectiveness of the treatment | [28] |
| ZVI FG CS | 1995 (10 years) | Monkstown, Ireland | The formation of a thin cemented layer at the PRB entrance, which was associated with the precipitation of Ca and Fe carbonates, crystalline and amorphous Fe sulfides and Fe (hydr)oxides | [73] |
| ZVI CT TCE Cr(VI) | 1996 (15 years) | Elizabeth City, North Carolina | The presence of mineral precipitates, such as iron oxides and sulfides, which did not significantly alter the hydraulic conductivity of the barrier | [29] |
| PTZ (pea gravel)—ZVI FG CS | 1996 (4 years) | Lakewood, Colorado | Mineral accumulation, mostly localized on the surfaces of iron particles collected near the upgradient aquifer-iron interface | [76] |

**Table 2.** *Cont.*

| Description | Year Installed (Monitoring Years) | Site | Observations | References |
|---|---|---|---|---|
| ZVI FG U(VI) | 1997 (10 years) | Fry Canyon, Utah | Groundwater velocity decreased approximately threefold due to the formation of mineral precipitates | [13] |
| ZVI CT U, NO$_3^-$ | 1997 (5 years) | Oak Ridge, Tennessee | Reduction in permeability and the consequent circumvention around cemented areas | [74] |
| ZVI CT CS | 1998 (>2 years) | Copenhagen, Denmark | Reduction in the hydraulic conductivity and circumvention of the barrier by approximately 1/5 of the contaminated plume | [71] |
| PTZ—ZVI FG As, Mo Se, U, V | 1999 (>5 years) | Monticello, Utah | Calcite mineralization was evident throughout the PRB but the contaminants were confined to the PTZ, which was composed of gravel and ZVI (13 % in volume); no hardpan was encountered in the PRB, indicating that calcium carbonate had not completely cemented any portions of the PRB | [70] |
| ZVI FG Mo, U | 2000 (>4 years) | Canon City, Colorado | Reduction in hydraulic conductivity after 2 years due to the precipitation phenomena observed at the barrier entrance | [13] |
| Calcite, vegetable compost, ZVI and sewage sludge CT Acid mine drainage | 2000 (3 years) | Aznalcóllar, Spain | The inefficient capture of the contaminated plume due to the improper PRB design; preferential flows within the PRB were due to the heterogeneities of the filling material | [72] |
| ZVI/sand (30:70 w.r.) CT Explosives | 2003 (>1 year) | Cornhusker, Nebraska | Reduction in permeability at the entrance of the barrier one year after installation, which was linked to an excess of biological activity or the incomplete degradation of the guaro rubber used during installation; the presence of sulfides and iron carbonates | [13,75] |

Note: w.r., weight ratio; CT, continuous trench; FG, funnel and gate; PTZ, pretreatment zone; CS, chlorinated solvents; TCE, trichloroethylene.

The cases summarized in Table 2 highlight the different profiles of the hydraulic behavior of PRBs and the different reasons that can cause reductions in permeability. This shows the complexity of the phenomenon and the need for further research in this field. To investigate the possible causes of reductions in the hydraulic conductivity of PRBs, it is necessary to understand the design parameters of PRBs, the installation techniques used and the data collected during the monitoring phases. The lack of these data makes it difficult to analyze the hydraulic behavior of PRBs. For example, knowing the geotechnical and hydrogeological characteristics of the aquifer in which the PRB is installed and the grain size distribution of the reactive media (or reactive medium) allows us to understand whether the PRB complies with the filter design criteria. Additionally, knowing the chemical composition of the contaminated groundwater flowing through the barrier and the physical and chemical properties of the reactive media, as well as the constant monitoring of the

hydraulic grade line upstream and downstream of the PRB, allows us to understand the causes of any reductions in the hydraulic conductivity of the barrier.

Data derived from PRB monitoring are even more necessary when the data are extrapolated from laboratory tests in which the behavior of PRBs is simulated on a small scale. In such cases, it is easy to control the boundary conditions and the test times are much shorter than real-world remediation times. This often involves the need to use prediction models, which strongly depend on the reliability of the conducted laboratory tests and the correct model calibration. The accuracy of these prediction models can be verified by means of data derived from PRB monitoring.

## 4. Physical and Mathematical Modeling of ZVI-Based PRBs

Physical and/or mathematical models are useful tools for understanding PRB behavior by correctly reproducing their operation. The physical model that is most commonly used at the laboratory scale is a one-dimensional pilot plant (column), in which the interaction processes between the aquifer and the reactive medium are simulated. On the other hand, mathematical models allow us to define the most suitable location and configuration of barriers and can be also used to predict the hydraulic and reactive behavior of PRBs in the long term [77–79]. A description of these models and the main results obtained regarding long-term ZVI hydraulic behavior are summarized in the following paragraphs.

### 4.1. Laboratory Experiments

A column test employs a cylindrical reactor, usually composed of Plexiglas [80,81], that is filled with a reactive medium. The flow of contaminated solution through the reactive medium, usually from bottom to top, is generally achieved using a pump. Along the entire height of the column, there are sampling ports from which it is possible to withdraw samples of the contaminated solution, usually using a needle that reaches the axis of the column. Then, different thicknesses of the material (or different residence times) and the ability of the reactive medium to remove contaminants can be evaluated. During column tests, it is necessary to evaluate hydraulic behavior over time using pressure transducers [63] or permeability tests [11]. Figure 4 shows a schematic diagram of an example of a column test apparatus.

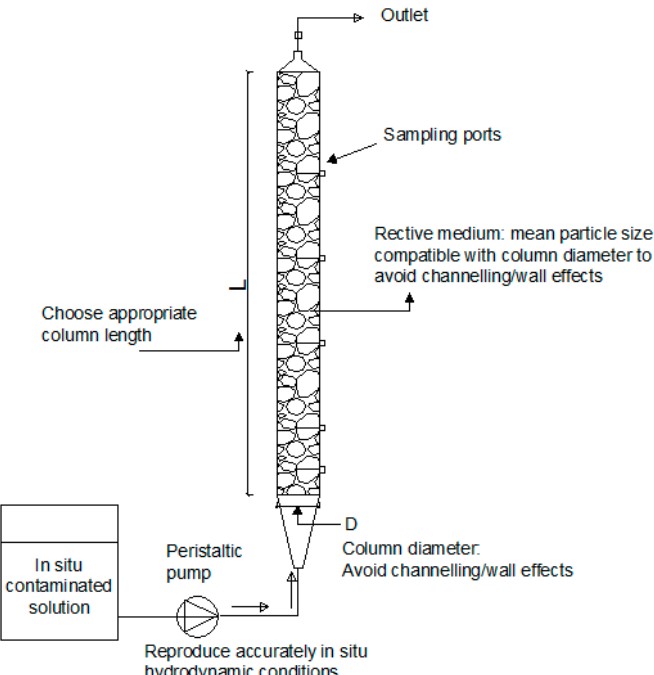

**Figure 4.** An example of a column test apparatus.

The two main issues associated with column tests are the possible channeling/wall effects and the accurate reproduction of in situ hydrodynamic conditions. The prevention of channeling and wall effects ensures that a small column of reactive medium behaves similarly to a large real-world PRB, particularly in terms of ensuring that the behavior of the reactive medium is not influenced by the proximity of the column surface. According to Badruzzaman and Westerhoff (2006) [82], these effects can be avoided if the ratio between the column diameter (D) and the mean particle size (d) is greater than 50. The reproduction of hydrodynamic conditions is somewhat more complex in practice. The best way to meet this criterion [83] is to keep the experimental flow rates and Peclet numbers (*Pe*) (Equation (1)) as similar as possible to the field values.

$$P_e = \frac{v_y d}{D_c} \tag{1}$$

where $v_y$ is the groundwater velocity in the longitudinal direction (ms$^{-1}$), $d$ is the mean or effective grain diameter (m) and $D_c$ is the diffusion coefficient of the contaminant in an aqueous environment (m$^2$s$^{-1}$) [84].

Moreover, the column length should be of the same order of magnitude as that of the filtration path in the full-scale technology in order to reproduce real operating conditions. This criterion is fairly easy to meet if the column length is appropriate; however, for the same reasons, rapid small-scale column tests that employ very short columns are not suitable for testing ZVI-based systems [38].

In addition, when employed for design purposes, the tests must be carried out using the flow velocity that was determined in situ and possibly groundwater that was withdrawn from the contaminated aquifer [13].

To investigate the long-term behavior of PRBs, for purposes other than design, it is possible to promote the aging of reactive media by conducting tests with higher flow rates, so that a significant volume of contaminated solution that is representative of several years of operation flows through the column in a short time. Before accelerating column tests, the reaction kinetics must be determined using the expected in situ flow rate; subsequently, the flow can be accelerated to age the active medium and then the real flow conditions can be simulated again to determine the long-term reaction kinetics and permeability [85]. However, it should be noted that the aging obtained by increasing the flow rate at the inlet to a column cannot exactly simulate the aging conditions that would occur in situ due to the different hydraulic residence times [43,45,85]. In particular, from previous accelerated column studies, it has been observed that PRB performance can be overestimated in terms of the long-term preservation of hydraulic conductivity [43]. In fact, the preservation of the hydraulic conductivity of granular mixtures of ZVI and lapillus has been observed because mineral precipitates and iron corrosion products do not accumulate at the entrance of the reactive medium and are instead more easily distributed over distances that are longer than those that normally occur in normal low-velocity conditions [11,86]. Therefore, the reliability of results obtained using the accelerated aging of reactive media must be carefully evaluated.

The data derived from column tests are usually depicted in terms of the normalized concentrations of contaminants, either as a function of the residence time or the duration of the test. The first possibility consists of plotting the normalized concentration of the contaminant at a given time as a function of the residence time (calculated for each sampling port). Data are often interpolated using a first-order kinetic equation and, in this case, it should be possible to derive the first-order kinetic constant (k). Following this approach, after determining the k value, the residence time that is necessary to achieve the regulatory limit can be found, starting from the initial concentration value. Finally, the minimum thickness of the PRB can be derived knowing the effective in situ flow velocity. However, many studies [87,88] have suggested that the real thickness of the PRB calculated using this deterministic design model should be increased by multiplying the obtained value by an appropriate safety factor (SF). This SF should also be used to take into account

several issues, such as the incomplete characterization of aquifers, seasonal variations and differences (in terms of flow rate and water composition) between column systems and real PRBs [16]. According to a modeling study conducted by Elder et al. [89], the *FS* value depends on the heterogeneity levels of the aquifer and can reach values that are even greater than 10 for highly heterogeneous aquifers.

Moraci et al. [49,90] reported examples of data interpretation using the first-order kinetic equation with reference to ZVI and granular mixtures of ZVI/pumice. The results of these studies clearly showed the reduction in the kinetic constant over time due to the reduction in the granular mixture efficiency. Under the hypothesis of constant hydraulic conductivity, the barrier thickness that is required to reach the target concentration should linearly increase over the designed life span.

The second mode that is often used for column data interpretation consists of plotting a graph of the normalized concentration of the contaminant at a given sampling port or a given thickness of the active medium as a function of time (Figure 5). From this curve, it is possible to identify the breakthrough time ($T_b$) and determine the removal capacity of the reactive medium (*RC*). $T_b$ is the time at which a clear and rapid increase in the contaminant concentration is observed (Figure 6). *RC* is defined as the ratio between the mass of contaminants removed at the breakthrough and the mass of the active medium [13]. This ratio must be assessed for reactive media that maintain the required hydraulic conductivity of PRBs over time. Following this second approach, under the hypothesis that contaminant concentration is constant with depth, the barrier thickness ($L_{PRB}$) can be calculated using the mass of the contaminant to be removed ($M_{contaminant}$), the barrier dimensions (i.e., the depth $H$ and width $W$) and the unit weight of the material ($\gamma$).

$$L_{PRB} = \frac{M_{contaminant}}{RC \cdot H \cdot W \cdot \gamma} \tag{2}$$

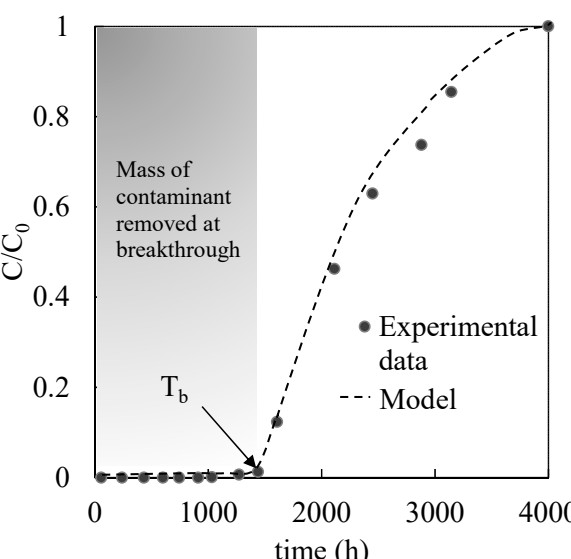

**Figure 5.** The expected breakthrough curve for a fixed sampling port.

The huge limitation of this expression is that the removal capacity is assumed to be constant with the thickness of the reactive medium. ZVI systems [11] have shown that the removal capacity does not remain constant along the thickness, and it has been hypothesized that the ZVI present in the part of a column that has not yet been affected by a contaminated plume is consumed in reactions with water and other constituents that are possibly present (e.g., nitrates), thereby decreasing its removal capacity.

Different mathematical models have been used in the literature to describe the breakthrough curves obtained using experimental data (Figure 5). The use of mathematical models allows us to (i) identify the possible reaction mechanisms involved in column

systems, (ii) predict the reactive behavior of materials and (iii) obtain useful parameters for technology designs [9,50].

The mathematical models that are generally used to simulate the adsorption capacity of granular media are the Thomas, Adams–Bohart and modified dose-response models [50,91–95]. These models are considered for calculating semianalytical approximation solutions to the one-dimensional transportation of the mass of a reactive dissolved contaminant in a saturated porous medium.

ZVI Hydraulic Behavior Studied Using Column Tests

Column tests are generally carried out with the objective of studying the reactive behavior of the tested material, and the hydraulic aspects are often neglected. Column studies that have examined hydraulic aspects are summarized in Table 3, including the reactive medium and the contaminants used in the column tests, the methods used to study the hydraulic behavior of the reactive medium and the main observations.

**Table 3.** Hydraulic behavior was investigated using column tests, which were carried out using reactive media composed of ZVI.

| Reactive Medium | Contaminant | Method | Observations | References |
|---|---|---|---|---|
| ZVI, ZVI mixed with sand | Chlorinated solvents | Pressure transducer and tracer test | Reductions in porosity occurred with water and no carbonates but high dissolved oxygen and did not occur with water and high carbonates and low dissolved oxygen | [63] |
| ZVI | Heavy metals and radionuclides | Tracer test | The major trapped gas, ($N_2$) affected permeability but not to the same extent as mineral precipitation, which was considered to be the primary mechanism for pore clogging around the inlet of the column | [60] * |
| ZVI | TCE + $CaCO_3$ | Manometer | The reduction in hydraulic conductivity was attributed to gas accumulation, precipitation did not appear to have a measurable effect on hydraulic conductivity | [96] |
| ZVI mixed with sand | Synthetic acid mine drainage (Al, Zn, Cd, Cu, Mn, Ni, Co, sulfates) | Tracer test | The reduction in porosity from an initial value of 0.55 to a final value of 0.39 was attributed to mineral precipitation | [97] |
| ZVI | $NO_3^-$ | Tracer test | The reductions in porosity of 25–30% were attributed more to mineral precipitation than trapped gases | [61] |
| ZVI | TCE TCE + $CaCO_3$ | Manometers | Gas production caused reductions in porosity of 10–20% (depending on the possibility of gas escaping from the column); the reductions in porosity caused by mineral precipitation varied from 14 to 36% (depending on the initial concentration of carbonates) | [98] |

**Table 3.** *Cont.*

| Reactive Medium | Contaminant | Method | Observations | References |
|---|---|---|---|---|
| ZVI | cis-DCE | Manometers | ZVI had a high corrosion rate in the presence of a high concentration of dissolved $CaCO_3$, which resulted in greater reductions in porosity near the influent face due to the accumulation of carbonate minerals | [64] |
| ZVI mixed with sand, gravel, pumice or anthracite | TCE | Tracer tests and gravimetric measurements | The reduction in porosity were attributed to gas accumulation and mineral precipitation, in the long term, gas accumulation in the pore spaces reduced due to microbial consumption | [99] |
| ZVI mixed with zeolite and activated carbon | Leachate | Constant-head permeability test | Hydraulic conductivity decreased with increasing treatment time and ZVI content | [100] |
| ZVI | Ni Zn Cu, Ni, Zn | Falling-head or constant-head permeability test | Clogging at the entrance of the column, the extent of which was linked to the influent concentration of the contaminants and influent flow velocity | [43,49,101] |
| ZVI mixed with pumice or lapillus | Ni | Falling-head or constant-head permeability test | Reductions in the hydraulic conductivity of mixtures with the highest contents of ZVI per unit volume | [11,43,45] |
| ZVI mixed with lapillus | Cu, Ni, Zn | Falling-head or constant-head permeability test | Granular mixtures with higher iron contents showed proportionally higher removal rates but also greater reductions in hydraulic conductivity over time | [50] |
| ZVI | - | Pressure transducer | The hydraulic conductivity of two different sizes of ZVI particles decreased in both small- and large-scale experiments | [102] |

Note: cis-DCE, cisdichloroethene; * results obtained using in-field column tests.

In most cases, the hydraulic conductivity of the reactive material was derived using pressure measurements or permeability tests, whereas in other studies, porosity was calculated by means of tracer tests. The causes of reduction in hydraulic conductivity or porosity were determined by taking into account the parameters investigated during the tests (e.g., aqueous species removal, the observation of gas bubbles, gravimetric measurements or the mineralogical composition of solid samples extracted from columns) or theoretical considerations (e.g., chemical reactions or mass/volume balance equations). The results summarized in Table 3 show that the hydraulic conductivity of reactive media reduces over time and in the absence of contaminants [102] and that the main causes of this phenomenon are different according to different authors.

The column tests carried out by Moraci et al. (2016) and Mackenzie et al. (1999) [49,63] showed that dissolved oxygen in the influent groundwater could cause clogging mainly due to iron oxides rather than carbonate precipitates. Mackenzie et al. (1999) [63] also showed that this phenomenon was reduced by using larger iron particles or, even better, when larger iron particles were mixed with sand of a similar size. Since clogging has been

observed with high dissolved oxygen levels, this phenomenon should not be an issue for anoxic aquifers. Through mass balances on carbonate losses and assuming an anaerobic corrosion rate of 1 mmol $Fe^{2+}$/kg iron/day, the authors concluded that the reduction in porosity that was measured by the tracer tests was not entirely due to mineral precipitation. Although bubbles of gas entrapment in the iron columns were not observed, the reductions in porosity were attributed to the accumulation of hydrogen film on the iron surface.

Other authors attributed the reductions in the porosity of PRBs to gas formation [55,63,96] and concluded that gas venting could be necessary, especially for closed systems.

Gas accumulation was also observed in reactive media that were composed of ZVI and different admixing agents, such as sand, gravel, pumice and anthracite [99].

In column tests carried out using granular mixtures of ZVI/pumice and ZVI/lapillus, long-term hydraulic conductivity was found to be linked to the amount of ZVI per unit of volume and the boundary conditions that were adopted in the tests (i.e., flow rate, type of contaminant and initial contaminant concentration) [11,103].

From an in-depth analysis of several studies on ZVI/$H_2O$ systems, Hu et al. (2020) [22] concluded that the volumetric expansive nature of iron corrosion was the most important physical phenomenon that occurred in those systems. As already mentioned, this expansion occurs because ZVI corrosion produces (i) $H_2$ (which occupies a volume approximately 3100 times larger than the volume of the parent ZVI [26,104]) and (ii) solid oxides and hydroxides (each of which is at least twice as large in volume as the ZVI ($V_{oxide} > V_{iron}$)). The volume of iron corrosion products and the volume of iron expansion are schematized in Figure 6. Hu et al. (2020) [22] concluded that for each ZVI filter, the temporal production of both $H_2$ and oxides was decisive for the long-term efficiency and permeability of the system. The authors indicated that the most important feature to consider was the fact that corrosion rates are never linear.

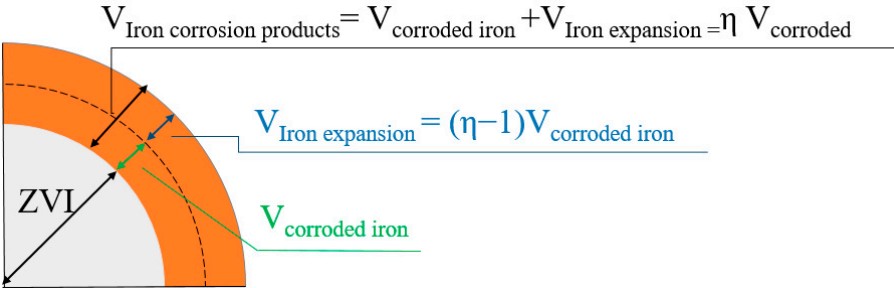

**Figure 6.** A representation of a ZVI particle, showing corroded iron volume, iron expansion volume and iron corrosion products.

For the design of ZVI-based PRBs, the possible reduction in the hydraulic conductivity of the barriers should be taken into account so that the long-term compatibility of the hydraulic conductivity of the PRBs with that of the base soils is assured and the permeability criterion of the granular filters is met.

### 4.2. Hydraulic and Geochemical Models

Mathematical models should mathematically reproduce the physical behavior of systems using a set of equations that take into account the boundary conditions. The application of mathematical models for PRB modeling requires the accurate geotechnical, chemical and hydrogeological characterization of the contaminated site and the PRB. The description of the reactive transport is very complex since it originates from the coupling of geochemical and hydraulic phenomena [105]. While the hydraulic phenomena concern the transportation processes of contaminants, the geochemical phenomena concern the behavior of chemical species when the chemical-physical conditions of systems vary [74]. The combination of geochemical and hydraulic models allows for the production of (more or less) simplified descriptions of real reactive phenomena. The hydraulic phenomenon can be described using well-established numerical models based on sets of partial differential

equations [1]. These models have been implemented in several commercially available software packages, including MODFLOW (a finite difference numerical method [106] that describes groundwater flow) and RT3D (which allows for the simulation of the reactive transportation of dissolved contaminants in groundwater using results obtained from MODFLOW as input). For the modeling of geochemical processes, it is possible to use equilibrium and kinetic models [107]. An example of an equilibrium model is MINTEQA2 [108], which is used to estimate the aqueous species and solid phases that exist in a state of thermodynamic equilibrium; however, this model does not account for the kinetics of reactions, i.e., the rates at which equilibrium is attained, and does not generally consider groundwater movement.

Kinetic models are appropriate when the flow velocity is non-negligible with respect to the reaction rates. They also include the effects of the spatiotemporal distributions of the reactions [109].

Hydraulic models can be coupled with geochemical models. For example, reactive transportation codes that incorporate kinetic geochemical algorithms to simulate contaminant degradation and mineral precipitation in PRBs have been used (e.g., using MIN3P [110,111] and RT3D [112]).

MIN3P is a multicomponent reactive transportation model for variably saturated porous media [113]. The model formulation is based on the global implicit solution approach, which allows for the investigation of interactions between chemical reactions and transportation processes in systems that are characterized by solid, liquid and gaseous phases. The hydraulic conductivity in the model was updated based on a normalized version of the Kozeny–Carman relationship. This software was initially used to establish the causes of reductions in iron reactivity over time so as to predict the long-term reactive behavior observed in column tests [111,114]. Subsequently, the software was modified to model the hydraulic behavior of reactive media, considering secondary mineral precipitation and gas formation [115]. Indraratna et al. (2014) [105] used commercial MODFLOW and RT3D commercial software to investigate the hydraulic behavior of a PRB composed of recycled concrete aggregates. The reduction in porosity due to the precipitation of secondary minerals is usually calculated using the volume of the minerals or stoichiometric calculations that consider variations in the aqueous concentrations of the elements involved in the reactions.

With the aim of modeling the performance of two PRBs, one filled with recycled concrete aggregates and one filled with limestone aggregates, Medawela and Indraratna (2020) [79] introduced a novel computational approach that coupled conventional geohydraulics with time-dependent changes in geochemical and biological parameters. Their model used commercial MODFLOW and RT3D software and took into account biological clogging.

### ZVI Hydraulic Behavior Studied Using Column Tests

The scientific literature has illustrated some examples of mathematical models that can predict both the hydraulic behavior of ZVI-based PRBs and identify the main causes of reductions in hydraulic conductivity. The results of these studies have reached different conclusions over the years. Some of these studies are summarized (in chronological order) in Table 4, which also shows results from using different mathematical models for different contamination contexts (i.e., organic or inorganic) and different causes of hydraulic conductivity reduction (e.g., mineral precipitation, gas accumulation or the expansive nature of solid iron corrosion products). Table 4 presents the details of parametric studies (PS) that aimed to identify the possible causes of reductions in the hydraulic conductivity of barriers and other studies that aimed to model the reductions in hydraulic conductivity that were observed in column tests (CTM). When used with reliable experimental results, this type of modeling is of fundamental importance for calibrating mathematical models that are used as forecasting models for PRB designs.

**Table 4.** The results obtained using mathematical models to predict the hydraulic behavior of PRBs.

| Reactive Medium-Permeating Solution | Model | Factors | Observations | References |
|---|---|---|---|---|
| ZVI-Natural ground water | MODFLOW and RT3D (PS) | Mineral precipitates | Porosity and hydraulic conductivity decreased over time; little change in hydraulic behavior over the 10 years following installation, but significant changes were expected after ~30 years, the magnitude of which was greatly influenced by the rate of major ions entering the PRB via advection | [116] |
| ZVI-Natural ground water | MODFLOW and RT3D (PS) | Mineral precipitates | The reductions in porosity were sensitive to the influent concentrations of $HCO_3^-$, $Ca^{2+}$, $CO_3^{2-}$ and dissolved oxygen, the anaerobic iron corrosion rate and the rates of $CaCO_3$ and $FeCO_3$ formation | [112] |
| ZVI-Chlorinated solvents | MIN3P (CTM) | Mineral precipitates | The reductions in porosity at the entrance of the reactive medium were due to the accumulation of carbonates, especially in the case of ZVI with a high degree of corrosion | [64] |
| ZVI-Chlorinated solvents | MIN3P (CTM) | Mineral precipitates and gas | The reductions in porosity were more related to the formation of gas than mineral precipitates | [115] |
| ZVI, ZVI/sand or pumice-Heavy metals | Kozeny–Carman Equation (CTM) | ZVI expansion | Assuming uniform corrosion, permeability decreased at the beginning of the filtration process as a consequence of the pores being filled with expansive iron corrosion products | [58] |
| ZVI-Heavy metals | Numerical -probabilistic model (CTM) | Contaminant precipitation, ZVI expansion and gas | The volumetric expansion of iron and mineral precipitation phenomena contributed to changes in the geometry of the pores of the reactive medium, determining a possible stop of generated gas bubbles; assuming the absence of gas (or its possible complete escape), higher values of iron corrosion rate were considered in order to fit experimental data | [49] |
| ZVI-water | Non-dimensional analysis (CTM) | Mineral precipitate | The exact cause of the reductions in permeability was irrelevant as the method proposed by the authors was general and could be applied to analyze permeability reductions | [102] |

Note: PS, parametric study; CTM, column test modeling.

The first studies only considered mineral precipitation as the possible cause of PRB clogging and supported the idea that the reductions in the porosity of ZVI-based PRBs were strictly linked to the geochemical conditions of the aquifers and the concentrations of $HCO_3^-$, $Ca^{2+}$ and $CO_3^{2-}$ ions [112,116]. Subsequently, when gas formation was then considered an additional cause of reductions in the porosity of ZVI-based PRBs, contrasting views emerged. In particular, when gas production and mineral precipitation were both considered, reductions in hydraulic conductivity were, in some cases, mostly attributed to gas formation [55,96,115], rather than mineral precipitation [60,61]. This mismatch could mostly be attributable to the possible differences in the extent of solids and gas formation due to the dissimilar conditions (e.g., flow rate, the chemical composition of groundwater and the oxic/anoxic conditions) of investigated systems [49] or, more likely, because the expansive nature of iron corrosion products was not considered as a cause of hydraulic conductivity reduction.

The first attempt to predict the time-dependent decrease in hydraulic conductivity on the basis of the volumetric expansion of corroding iron was carried out by Bilardi et al. (2013) [58] through the use of the Kozeny–Carman equation. Subsequently, this phenomenon was considered in addition to mineral precipitation and gas formation using a numerical probabilistic model [49]. The results of this study highlighted the fact that the iron corrosion rate adopted in the model strongly influenced the hydraulic behavior of the system.

As suggested by Hu et al. (2020) [22], predictive models should take into account the nonlinear kinetics of iron corrosion rates, for these reasons, long-term investigation into iron corrosion rates in ZVI-based systems that simulate PRB operation is required.

## 5. Strategies to Improve the Hydraulic Behavior of ZVI-Based PRBs

Due to the possible reductions in the hydraulic conductivity of ZVI, different strategies have been proposed in the literature. Among these, the most common strategy is to mix ZVI with granular materials. This strategy has been considered as a possible method to improve the hydraulic and reactive behavior of PRBs [51,58,81,117]. ZVI can be mixed with inert granular materials, such as sand or gravel, and/or active double porosity granular materials, such as pumice or lapillus [11,43,63,81,99,118].

Mixing ZVI with reactive or inert materials allows the ZVI to disperse throughout larger volumes, thereby preventing the clogging phenomenon while also increasing the contact time between the contaminated solution and the reactive medium. In some cases, this has improved the removal efficiency [11,103] (Figure 7). Moreover, admixing ZVI can reduce costs [58] and avoid material waste [119].

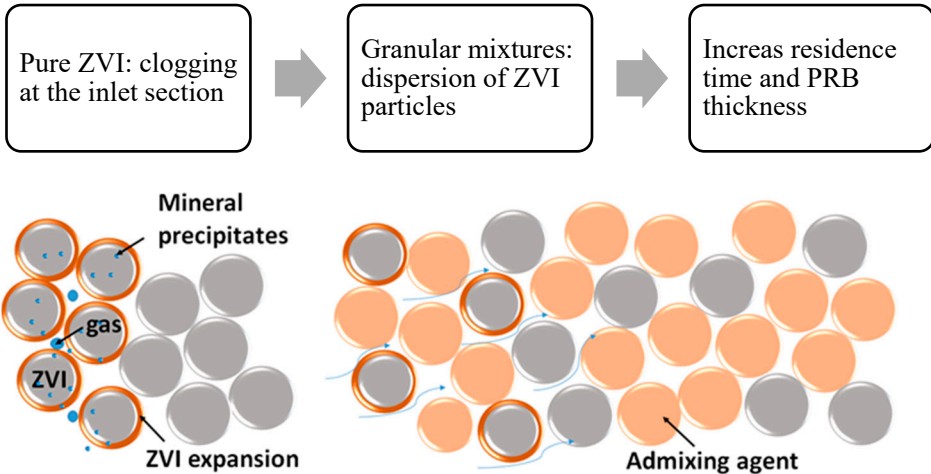

**Figure 7.** A schematic representation of the clogging phenomenon in pure ZVI and ZVI dispersed in an admixing agent.

Komnitsas (2007) [80] studied the copper removal efficiency of ZVI/sand mixtures. The hydraulic behavior was not investigated, but SEM images revealed the bulbous formation of a mixed iron (oxy)hydroxides layer.

Jun et al. (2009) [120] investigated a granular mixture of ZVI and zeolite for the remediation of landfill leachate-polluted groundwater. They found that the pollutant removal efficiency of the mixed media was higher than that of the ZVI alone (the hydraulic behavior was not studied).

A pilot PRB composed of ZVI mixed with leaf compost, limestone and pea gravel was installed in Charleston (SC), with the aim of removing heavy metals and arsenic from groundwater [121]. The slight reduction in hydraulic conductivity in the lower half (bottom) of the PRB after 18 months could be indicative of precipitation-induced clogging gradually occurring within the PRB.

Moraci and Calabrò (2010) [81] introduced granular mixtures of ZVI/pumice for the removal of heavy metals, such as copper and nickel. These mixtures were able to optimize the use of ZVI by increasing the groundwater residence time in the barrier, thereby achieving an adequate level of treatment.

Ruhl et al. (2012) [99] tested sand, gravel, porous pumice and anthracite combined with ZVI for the remediation of TCE-contaminated groundwater. The four considered support materials revealed different hydraulic properties but achieved comparable results with regard to reactivity and effluent concentrations. However, for long-term operation, PRBs consisting of ZVI mixed with porous materials with large pores were found to better retain the hydraulic conductivity in the long term.

Bilardi et al. (2013) [58] indicated that admixing ZVI with sand and pumice resulted in the extended service life of the barrier for the treatment of groundwater contaminated by copper, nickel and zinc. In that study, the longest service life was observed for a system with pumice as the admixing agent, which was consistent with the fact that the intraparticle porosity of the material helped to avoid permeability reductions. The authors of [51] also stated that the ZVI proportion in efficient real-world systems should be <50% (1:1, *v/v*).

According to Calabrò et al. (2012) [118], Madaffari et al. (2017) [11] and Bilardi et al. (2020) [103], the optimum choice of ZVI per unit volume (or the composition of the granular mixture) when using pumice or lapillus is strictly dependent on the flow velocity of the groundwater through the PRB, the type of contamination and the initial concentration of contaminants. In particular, by reducing the initial contaminant concentration and/or the flow velocity, the ZVI dispersion rate should increase (i.e., a ZVI content of $\leq 30\%$ by weight should usually be adopted) in order that the contaminant removal does not occur in the first few centimeters of the reactive medium and the risk of hydraulic conductivity reduction is reduced.

Zhou et al. (2014) [100] tested mixtures that were composed of ZVI, zeolite and activated carbon (AC) to remediate groundwater that was heavily contaminated by landfill leachate. The most effective weight ratio of the ZVI/zeolite/AC mixture was found to be 5:1:4, based on the observed reactive and hydraulic behavior.

Han et al. (2016) [122] investigated the removal efficiency of different heavy metal ions by acid-washed ZVI and zerovalent aluminum (ZVAl) in PRBs. They found that the reactive performance of columns filled with a mixture of acid-washed ZVI and ZVAl was much better than that of columns filled with ZVI or ZVAl alone.

Madaffari et al. (2017) [11] showed that lapillus was a suitable admixing agent for ZVI (as is pumice), since it allowed for both the optimization of the use of ZVI and the preservation of its hydraulic conductivity for the treatment of nickel-contaminated solutions. Based on their results regarding ZVI mixed with lapillus at three different weight ratios (i.e., 10:90, 30:70 and 50:50), the design parameters of PRBs (i.e., filter width, ZVI/lapillus weight ratio and the total mass of ZVI) could be adjusted according to the expected flow velocity and the concentrations of contaminants.

Bilardi et al. (2020) [103] compared the performance of ZVI mixed with two volcanic materials (i.e., lapillus and pumice (70% in weight)). The experimental results showed that

the ZVI/lapillus mixture had the best zinc and nickel removal efficiency. This behavior was linked to the non-negligible removal capacity of both heavy metals by lapillus, which was confirmed by batch tests.

An important geotechnical aspect that has often been neglected in the literature is the evaluation of the grain size distribution (GSD) curve of each material when two or more granular materials are mixed in order to derive an internally stable mixture [45]. Internally unstable soils are those characterized by a concave upward GSD curve, gaps inside the GSD curve (gap-graded soils) or broadly graded GSD curves (Figure 8) [123]. Furthermore, mixing iron particles with an admixing agent that has smaller particles causes reductions in initial porosity and does not allow for the iron particles to properly separate, thereby causing cementation during the corrosion process.

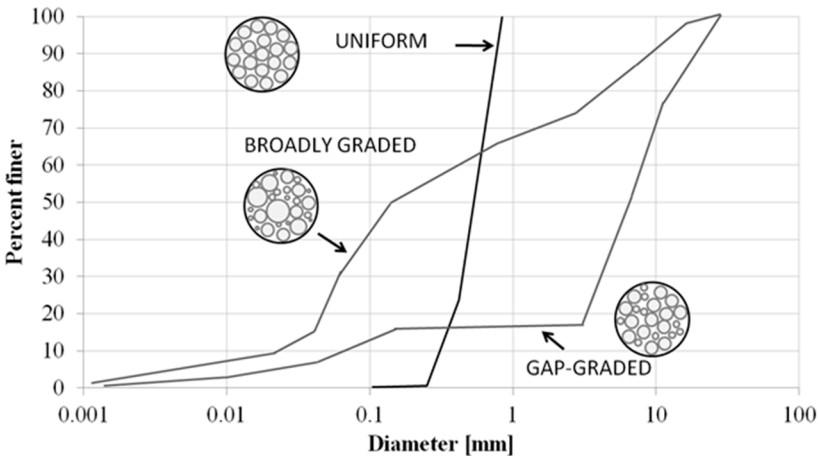

**Figure 8.** The typical grain size distributions of internally unstable (broadly graded and gap-graded soils) and stable soils (uniform soils).

Moreover, when ZVI is used in granular mixtures, it is advisable to refer to volumetric ratios rather than weight ratios in order to obtain a greater awareness of iron dispersion [38]. This is because the ZVI specific gravity is usually considerably greater than that of the supported material (i.e., pumice or granular activated carbon).

Another possible way to improve the hydraulic behavior of ZVI barriers is the use of sacrificial pretreatment zones (PTZs) upstream of the barrier, which consist of mixtures of ZVI and coarse inert granular materials (i.e., gravel or sand) that are characterized by the bland presence of iron. This region creates an area in which the pH of groundwater is higher, oxygen eventually becomes present and mineral-forming ions are consumed by the reaction with ZVI in order to reduce the precipitation of secondary minerals within the most reactive inner area designed for contaminant removal [70,124]. This strategy was studied by Li and Benson (2010) [109] by means of a numerical model coupled with a groundwater flow model (MODFLOW) and a reactive transportation model (RT3D). The authors highlighted that although this solution limited reductions in porosity within the reactive zones due to the lower precipitation of secondary minerals, the pretreatment zones did not eliminate reductions in porosity completely because secondary minerals (e.g., $Fe(OH)_2$) still formed within the reactive zones in response to iron corrosion. This configuration requires further study through suitable experimental tests.

When the clogging of reactive media at PRB entrances is expected, it is also possible to place the reactive media in zones that are easily accessible for substitution. This method was adopted in a ZVI-based PRB field application for a small and shallow contamination plume [73]. In this particular funnel and gate configuration, the reactive medium was placed in a circular reaction vessel (1.2 m in diameter), which was filled at the bottom with the reactive medium but was empty in the middle and upper sections. Access to the vessel was possible through a manhole on the surface, and gases were removed from the PRB via a modified streetlamp. The flow moved vertically downward, so any mineral precipitation

took place at the entrance (top) of the iron column, which could easily be removed when a substantial reduction in flow was observed. Moreover, a ventilation pipe was installed to disperse the gases into the atmosphere and prevent the buildup of gases. This configuration helped to reduce the buildup of precipitates over 10 years of operation [73].

## 6. Design Steps for Hydraulically Efficient ZVI-Based PRBs

ZVI mixed with a nonexpansive granular material is the most suitable choice for long-term hydraulically efficient PRBs for the remediation of heavy metal-contaminated aquifers. Unfortunately, the hydraulic aspects are often neglected during the design and monitoring of the operation of PRBs. For the reasons explained in this review, a procedure for the design of PRBs composed of granular mixtures containing ZVI is summarized in the following steps (also schematized in Figure 9):

1.  Conduct a detailed characterization of the site to accurately determine the extent, type and concentration of each contaminant present in the aquifer, the geotechnical characteristics of the soil in the contaminated aquifer and the hydrogeological characteristics of the aquifer.
2.  Select the possible ZVI grain size based on the grain size distribution of the soil constituting the aquifer.
3.  Select the optimum admixing agents, which should have a similar grain size distribution to that of ZVI.
4.  Select the optimum reactive medium through batch tests.
5.  If the granular mixture is reactive to contaminants, select the optimum volumetric ratio and carry out column tests to define the optimum thickness of the barrier and assess the long-term removal efficiency and the long-term trends of the hydraulic conductivity of the reactive medium. If clogging occurs, a more dispersed configuration should be tested.
6.  Define the barrier configuration based on proper numerical modeling.
7.  Determine the specifications of the materials and construction methods and define a detailed control and monitoring plan.

Regarding step 5, in order to clarify the long-term performance of ZVI systems, it is important to associate the hydraulic and reactive behavior observed in long-term column tests with the physical characterization of the ZVI and the assessment of its intrinsic reactivity and long-term corrosion rate.

Regarding step 6, Pathirage and Indraratna (2014) [78] proposed a useful flowchart to determine the optimum width of PRBs by means of iterative simulations carried out using MODFLOW coupled with RT3D.

Finally, with reference to the last step, the monitoring phase is essential for evaluating the correct operation of the PRBs. In particular, at least the contaminant removal efficiency needs to be assessed, along with the possible formation of intermediate reaction products, any changes in the quality of the groundwater, groundwater flow (especially the possible circumvention of the barrier) and any variations in hydraulic conductivity over time. These data are essential for evaluating the efficiency of PRBs over time and anticipating any necessary replacements of the reactive material or other maintenance operations.

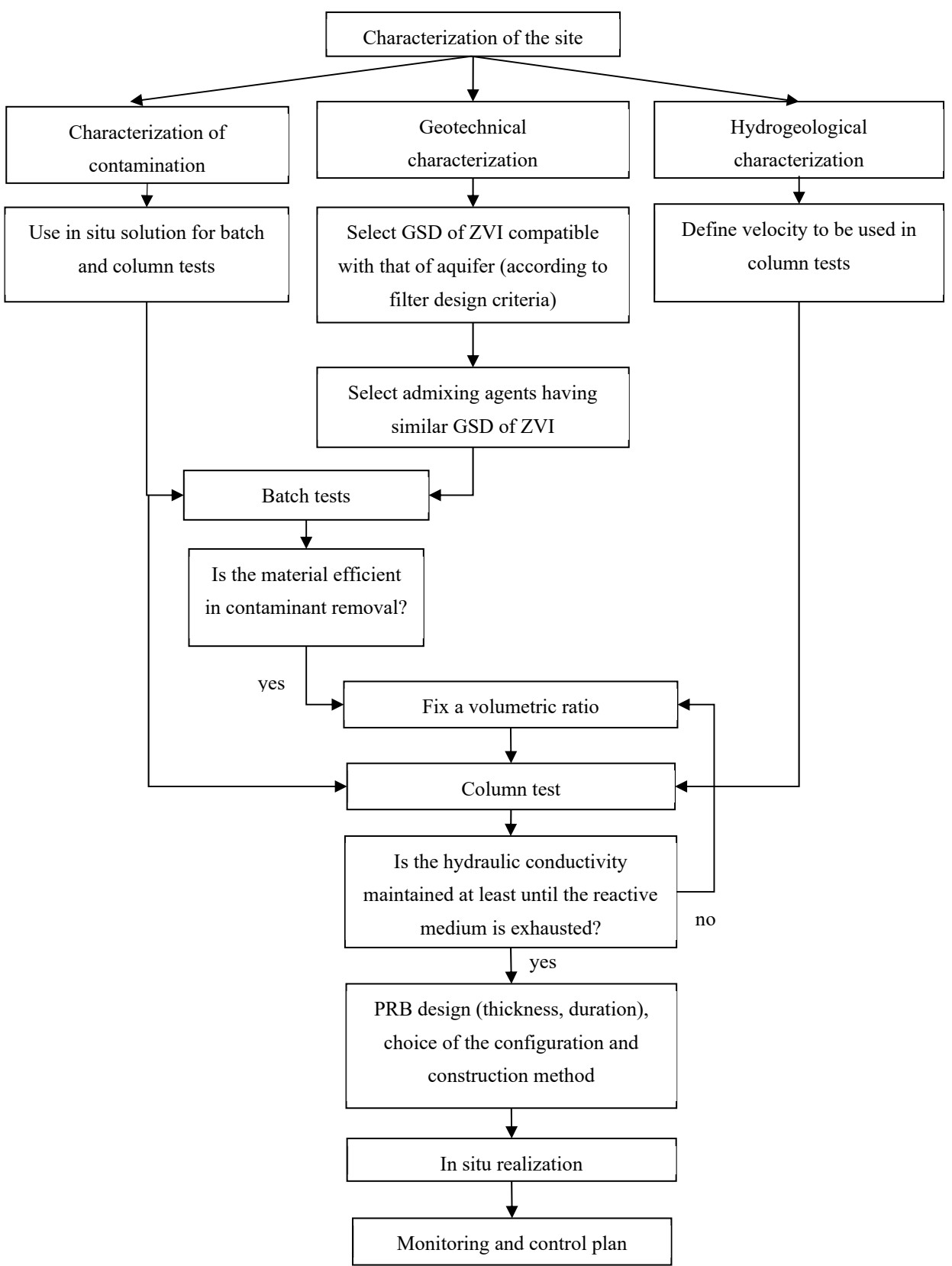

**Figure 9.** A flowchart of the design of hydraulically efficient ZVI-based PRBs.

## 7. Conclusions and Suggestions

If properly designed, PRBs represent a sustainable remediation technology since they can treat contaminated groundwater without energy consumption, exploiting natural

hydraulic gradients or wasting material, as long as the reactive media are able to reach the long-term remediation targets without being replaced.

This review summarizes studies on the hydraulic behavior of ZVI-based PRBs in field applications, long-term column experiments and numerical modeling. Processes related to iron corrosion (i.e., expansion and solid and gas formation) and mineral precipitation can influence the permeability of PRBs, especially at the aquifer–PRB interface, and the extent of this phenomenon depends on different factors, such as groundwater velocity, geochemical conditions and the contamination and grain size distribution of the reactive medium.

Mixing ZVI with inert or reactive media represents a good strategy to avoid the possible clogging of the reactive media at PRB entrances. The dispersion of ZVI in porous granular mixtures, such as pumice or lapillus, increases the barrier thickness and reduces the clogging phenomena related to iron corrosion. The choice of the degree of iron dispersion inside the granular mixture so as to ensure a good compromise between reactivity and hydraulic conductivity is strictly linked to groundwater velocity, the type and concentrations of contaminants and the geochemical characteristics of the aquifer. Therefore, iron dispersion should be adjusted according to the two possible limiting states of barriers: reductions in intrinsic reactivity (which obstruct target removal) and reductions in hydraulic conductivity (which obstruct aquifer flow through barriers). High pollution loads or high groundwater velocities could require lower iron dispersion rates and vice versa (i.e., lower pollution loads or lower groundwater velocities could require more dispersed configurations of iron particles). These configurations have to be established by means of long-term column tests carried out simulating the in situ conditions.

The main issue that remains to be solved is the difficulty in understanding the long-term behavior of ZVI-based PRBs, which is related to the iron corrosion rate and its evolution over time. In the last few years, several researchers have implemented more and more advanced numerical models to simulate the long-term hydraulic behavior of barriers in a more realistic way. They could represent important tools for defining the best configurations of PRBs, as well as their location and orientation. The application of numerical models able to simultaneously describe the hydraulic and reactive behavior of PRBs and supported by experimental data derived from field-scale systems or long-term column experiments could help us better understand the operational limiting states of barriers.

**Author Contributions:** Conceptualization, S.B., P.S.C. and N.M.; methodology, S.B., P.S.C. and N.M.; investigation, S.B.; writing—review and editing, S.B., P.S.C. and N.M.; visualization, S.B.; supervision, P.S.C. and N.M. All authors have read and agreed to the published version of the manuscript.

**Funding:** This research received no external funding.

**Data Availability Statement:** The data presented in this study are available within the article. For the availability of detailed data sets the corresponding authors can be contacted.

**Conflicts of Interest:** The authors declare no conflict of interest.

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
