# Peer review of "A Review of the Hydraulic Performance of Permeable Reactive Barriers Based on Granular Zero Valent Iron"

_water, doi:10.3390/w15010200_

Round 1

Reviewer 1 Report

This paper reviews the recent advance of PRB with the use of ZVI. It shows value to this field. However, I have the following concerns.

(1) Abstract: the background is too long. Please focus on the results/findings of the paper.

(2) Introduction: Please introduce previous review papers on this issue and comment on the limitation of previous review papers. Besides, clarify what is the difference between your review and those and what is the unique novelty of  your review?

(3) What is the definition of hydraulic behavior of PRB? What indicators are included? How to judge good or bad hydraulic performance?

(4) There are two Section 6?

(5) English writing needs further polish through the whole text.

Reviewer 2 Report

The review article on 'the hydraulic performance of permeable reactive barriers based on granular zero valent iron' needs major corrections.

The article has been  written in a very casual and generalized manner and most of the literature studies are not well discussed. The paper also lacks references to many statements made in the paper.

Authors have not discussed any numeric values (optimum requirement based on their study) in the paper, only general statements are made for any studies reported. No critical analysis by authors has been provided.

The Tables are also not well represented.

The overall formatting (numberings, mentioning any literature discussed) of the paper has many mistakes.

My detailed comments can be found below.

1)    Provide a list of studies, showing the best conditions they determined for a PRB system.

2)    Rewrite the objective of the study in the introduction part. Mentioning the challenges faced during the designing of PRBs and how the paper helps in explaining these challenges.

3)    In section 2.1, line 104-109, provide some examples, supporting the statement. Provide some literature showing comparative study of different waste materials used in designing PRB and how their properties their selection criteria.

4)    In section 2.1 provide some literature studies supporting the argument made. Line 139-149.

5)    The section numbering is wrong. After 2.1, 2.1 is repeating.

6)    Provide some literature supports. Line 159 “a finer particle size….” Mention some range or values associated with PRBs. “Although this aspect……. In the literature” put reference and conclusion of these references.

7)    Table 1, mention some examples, materials that are preferred and supporting literatures.

8)    The manuscript lacks the support of literatures. Please modify the manuscript accordingly.

9)    Section 3 line 196-198. Put reference. Line 199-203, “the hydraulic conductivity……. Aquifer”. What are the general values used, put reference to those?

10)         Line 218 “laboratory studies…..” what are the materials studied. Mention those.

11)       Rewrite 3.1, mention the properties, discuss some recent literature studies supporting the argument.

12)       Line 279 “According to the study” please follow a proper format while mentioning a study.

13)       Line 300 “ The good performance of PRBs ….. operated by [62] and after 15 years [63]…..” please follow a correct format while mentioning any studies.

14)       Line 308 “the hydraulic performance of a PRB could also depend on construction method” no construction methods are discussed in this section. Discuss some studies from table 2 here.

15)       Line 404. Discuss about those studies. What is the outcome and how that affects the PRB property. How much thickness of PRBs is preferred in those studies? Mention.

16)       Line 690-697, give references.

17)       Mention some ratios of sand and ZVI mixing that are studied and preferred in ZVI-PRB systems.

18)       The review is very much generalized without any examples or studies taken in focus. Rewrite accordingly.

19)       Line 747, ref 72 proposed a flowchart. Discuss, what is the outcome of those steps which authors have considered while writing the paper. Line 750-754, mention the reason.

20)       Section 6, line 718 give references, to support the steps of granular mixing of ZVI. Check the numbering, section 6 is being repeated.

21)       While mentioning literatures studies and their outcomes, provide the data they suggest to be used and how it varies form the already preferred values.

The articles needs to address all the comments before acceptance. 

Round 2

Reviewer 1 Report

The abstract still needs major revision. As I proposed, the abstract should concentrate on your core findings of the review. The current version only covers the introduction of PRB, the aim of the review and what you do too generally. To be specific, I suggest the authors follow the structure: 1 sentence of introduction, 1 sentence of your aim, 1 sentence of what you do in this review, 5-6 sentences of your findings/opinions after reviewing these references, and lastly one sentence of outlooks/suggestions for future research. This is what a high-quality abstract should look like.
